# The Coronavirus Disease Ontology (CovidO)⋆

Sumit Sharma[1][0000−0001−5054−8670] and Sarika Jain[2][0000−0002−7432−8506]

Department of Computer Applications,
National Institute of Technology Kurukshetra, Haryana 136119, India
sharma24h@gmail.com, jasarika@nitkkr.ac.in

**Abstract.** This paper presents the Coronavirus Disease Ontology (CovidO), a superset of the available Coronavirus (Covid-19) ontologies, including all the possible dimensions. CovidO consists of an ontological network of thriving distinct dimensions for storing coronavirus information. CovidO has 175 classes, 169 properties, 4141 triples, 645 individuals with 264 nodes and 308 edges. CovidO is based on standard input of coronavirus disease data sources, activities, and related sources, which collects and validates records for decision-making used to set guidelines and recommend resources. We present CovidO to a growing community of artificial intelligence project developers as pure metadata and illustrate its importance, quality, and impact. The ontology developed in this work addresses grouping the existing ontologies to build a global data model.

**Keywords:** Coronavirus Ontology · Ontology Learning · Semantic Metadata · Semantic Web · Covid-19.

## 1 Introduction and Motivation

The World Health Organization declared a Public Health Emergency of International Concern on 30 January 2020 and a pandemic on 11 March 2020 [1]. At the same time, novel coronavirus (Covid-19) pandemic data has been collected by various data sources like the World Health Organization (WHO) and DXY.CN, BNO News, National Health Commission (NHC) of the People's Republic of China, China Centers for Disease Control and Prevention (CCDC), Hong Kong Health Department, Macau Government, Taiwan CDC, US Centers for Disease Control and Prevention (CDC), Government of Canada, Australian Government Department of Health, European Centers for Disease Prevention and Control (ECDC), Ministry of Health Singapore (MOH), and others. The Covid-19 data sources and access links are shown below.

– WHO https://www.who.in
– DXY.cn https://www.dxy.cn
– BNO News https://bnonews.com
– National Health Commission (NHC) http://en.nhc.gov.cn

---

⋆ Supported by NIT Kurukshetra, Haryana, India.

– China CDC (CCDC) https://www.chinacdc.cn/en/
– Hong Kong Health Department https://www.dh.gov.hk
– Macau Government https://www.gov.mo/en/
– Taiwan CDC https://www.cdc.gov.tw/En
– US CDC https://www.cdc.gov/
– Government of Canada https://www.canada.ca/en.html
– Australian Government Department of Health https://www.health.gov.au/
– ECDC https://www.ecdc.europa.eu/en
– Ministry of Health Singapore (MOH) https://www.moh.gov.sg/
– Ministry of Health and Family Welfare https://www.mohfw.gov.in/

A significant issue is that the data sources related to Covid-19 are heterogeneous, static, and broad in scope. So many heterogeneous and stationary data sources create situations where data is sometimes under-utilized or, in more extreme cases, not used for the decision-making process [2]. Decision-making impacts all business processes and human life. For example, Covid-19 cases increases are linked to the company's performance as risks to human health and safety. As a result, dealing with multidimensional (e.g., volume, variety, etc.) aspects of information is one of the primary difficulties in today's data management and processing environment. Another vital issue of Covid-19 is to provide semantic (machine understandable) representation of data from various exciting fields such as research, health, resources, drugs, and treatment. Ontology is emerging to solve these issues. Ontology solves the problem of changing user expectations and data integration demands driven by its volatility in a rapidly growing digital market and societal challenges related to resource efficiency [3]. Ontologies have proven practical tools for representing domain knowledge, integrating data from disparate sources, and supporting many semantic applications [4].

This paper presents an OWL-based Coronavirus disease ontology (CovidO) that defines all the possible concepts, features, attributes, and relations to describe Covid-19. To develop CovidO, we have defined seven dimensions that cover all essential aspects of Covid-19: (1) About the Covid-19 infectious disease, symptoms, drugs, and treatment; (2) Information statistics of Covid-19 cases in a geographical region; (3) Covid-19 patients information with the cause of infection and exposure of pandemic; (4) Covid-19 related resources and their availability in a location; (5) Impact of Covid-19 in different verticals like education, finance, business, research and social; (6) Various guidelines and prevention and vaccine mandates by public authority; (7) Global and biomedical research on Covid-19. Several ontologies (e.g., Infectious Disease Ontology (IDO), Virus Infectious Disease Ontology (VIDO), Coronavirus Infectious Disease Ontology (CIDO), etc.) have been created for coronavirus one after another. They comprehensively and thoroughly describe coronavirus disease. These ontologies individually fail to cater to all the dimensions of coronavirus. The ontology developed in this paper aims to group existing ontologies to construct a common global data model with the unified purpose. The main contributions of this paper are:

– to provide a list of dimensions to cover every aspect of coronavirus-related information.

- to develop standard metadata (Providing a schema) called CovidO as a global data model to annotate the Covid-19 information.

This paper presents our work on CovidO version 1.0.1, which consolidates Covid-19 concepts: https://w3id.org/covido. To be independent of external ontologies, we defined a new namespace https://w3id.org/covido with the prefix covido (registered entry at http://prefix.cc) for all classes used in the ontology. As a permanent URL service, we use w3id.org. The relevant conceptual design, competency question, ontology, data, and code are publicly available to the community through a GitHub project.

The contents of this paper are organized as follows. The related work describes literature on the various existing Covid-19 associated ontologies and their scope boundary. The dimension section describes the incidence of the Covid-19 information dimensions in the primary analysis and provides a summary analysis. The conceptual design and scope of CovidO are described in the CovidO section to build the ontology. The ontology design section defines the competency questions to determine the scope of CovidO with abstract design. The method section outlines the model to develop CovidO. In the results section, we present the main results for the Covid-19 schema, in addition to simple predictions for the future incidence of COVID-19. Some concluding remarks are given in the conclusion.

## 2   Related Work

Several ontologies represent the Covide-19 pandemic in different contexts. We have found 12 ontologies related to Covid-19, each representing a completely different scope of Covid-19. They are briefed here:

O1: **Infectious Disease Ontology (called IDO):** IDO [5] is an interoperable ontology that comprises domain information on an infectious illness, including elements relevant to the disease's clinical and biological aspects. The entity 'disease' is retrieved from the Ontology for General Medical Science (OGMS) [6] ontology and serves as a foundation of the IDO ontology.

O2: **Virus Infectious Disease Ontology (VIDO):** VIDO [7] adds virus-specific concepts to the IDO ontology and gives factual information on the domain of virus sickness. VIDO, in particular, inherits the entities from IDO by adding the word 'virus' to create a subclass, and the logical and textual information about the classes are updated accordingly. VIDO also uses elements from the Open Biomedical Ontologies (OBO) Foundry ontologies [8], such as SARS-CoV-2 proteins from the Protein Ontology.

O3: **Coronavirus Infectious Disease Ontology (CIDO):** CIDO [9] is a coronavirus-specific ontology, broader VIDO. CIDO introduces eight new entities from the coronavirus domain and concepts from other ontologies related to VIDO [7].

O4: **Covid-19 Infectious Disease Ontology (IDO-Covid-19):** IDO-Covid-19 [10] is the most particular version of CIDO, containing information

about Covid-19 and its cause SARS-CoV-2. IDO-Covid-19 adhers to the OBO Foundry design philosophy by extending the CIDO in the same way as the CIDO extends VIDO and the VIDO extends the IDO. IDO-Covid-19 also pulls concepts from other ontologies, such as SARS-CoV-2, imported from NCBITaxon.

O5: **COviD-19 Ontology for the case and patient information (called CODO):** CODO [12] is an ontology that contains COVID-19 case data in a format that can be used by other ontologies and software systems and is based only on OWL and different W3C standards. CODO tracks specific pandemic cases, including information such as how the patient is considered to have been infected and potential further contacts who may be at risk owing to their association with the infected individual. CODO also tracks clinical tests, travel history, available resources, actual demand (e.g., ICU bed, invasive ventilators), trend analyses, and forecast increases.

O6: **COVID-19 surveillance Ontology (COVID19):** COVID-19 surveillance Ontology [11] is a Covid-19 application ontology that intends to provide Covid-19 cases and respiratory information by obtaining data from multiple medical records systems. This ontology is constructed as a taxonomy with only 32 classes. COVID-19 verified by a lab test, SARS-CoV-2 identified, Probable Covid-19, Clinical codes, Possible COVID-19, Suspected COVID-19, Under investigation, Exposure, COVID-19 excluded are the ten core ideas of COVID19 ontology. The COVID19 ontology was created using the protégé tool, and its format is based on the OWL language.

O7: **Covid19-IBO:** The Covid19 Impact on Banking Ontology (Covid19-IBO) [13] is a knowledge graph that gives semantic information regarding Covid-19's influence on India's banking industry. In addition, the authors have provided a schema matching technique with satisfactory results for mapping the Covid-19 ontologies.

O8: **Kg-COVID-19:** The KG-COVID-19 [14] framework is used to create customized COVID-19 knowledge graphs. The FAIR (Findable, Accessible, Interoperable, and Reusable) approach is followed by KG-COVID-19, which combines various COVID-19-related data.

O9: **COVID-19 Ontology:** The COVID-19 ontology includes the function of molecular and cellular entities in viral-host interactions throughout the virus life cycle and a wide range of medical and epidemiological concepts associated with COVID-19. A scalable new coronavirus (SARS-CoV-2) entity is represented as an ontology. As a prominent target of ongoing COVID-19 medicinal research, the ontology contains a broad scope on chemical entities ideal for drug repurposing. The ontology's performance was evaluated using Medline and the Allen Institute's COVID-19 corpus.

O10: **DRUGS4COVID19:** DRUGS4COVID19 [15] identifies drugs and their associations with COVID-19. The ontology's core concepts include drug, effect, disease, symptoms, disorder, chemical substance, and so on.

O11: **COVIDCRFRAPID:** The World Health Organization's (WHO) COVID-CRFRAPID [16] ontology is a semantic data model for the COVID-19 RAPID case record form. COVIDCRFRAPID ontology provides semantic

references to the form filled by patients during the treatment as questions and responses. It shows a variety of application scenarios, including graph-based machine learning.

O12: **ROC:** Ontology (Country Responses towards COVID-19) ROC [17] enables data integration from heterogeneous data sources and answers interesting questions. ROC was designed to assist statistical analysis in exploring and analyzing the efficacy and side effects of government responses to COVID-19 in various nations.

We investigate these existing ontologies, focusing on a group of individuals to discuss a specific topic like drug, protein interaction databases, protein function annotations, Covid-19 patient, and cases. So they have a limited scope that does not cover all aspects of COVID-19. We have found that these ontologies refer to the Covid-19 pandemic but represent different aspects and scopes. We fill this gap, and our work follows the same approach to ontology design and has a common motivation. We have conglomerated all these ontologies into a comprehensive design covering all the required distinct dimensions (Covid-19 cases information, patient information, disease-symptom-treatment, resources, Covid-19 impact, research, and event or news related to Covid-19) discussed in the next section. By adopting established models, we aim to facilitate integration, linking, and reuse across the data sources and make data accessible to a wide range of applications. In addition, new entities have been introduced required.

## 3  Covid-19 Information Dimensions

With a view to allowing stakeholders in the research communityand application developers to reach out and benefit, CovidO has been created as a platform through specific dimensions. There are many domains and subdomains; we group these domains into seven significant divisions that cover all the aspects of Covid-19 related knowledge. These seven dimensions are shown in detail in Table 1. The CovidO ontology is developed based on these dimensions representing COVID-19 information in OWL format and other W3C standards utilized by other ontologies and software systems. The last column of Table 1 describes the core CovidO classes associated with a particular dimension.

CovidO allows detailed tracking of specific pandemic dimensions. For example, the diseases and treatment dimension includes clinical test tracking, test report history, illness, symptoms, medication, and clinical measurement and diagnosis. Similarly, CovidO traces other dimensions as well. In brief and with overall dimensions, CovidO monitors the Covid-19 patient's travel history, symptoms, medication, available health care facilities, resources, actual need (e.g., ambulance, invasive ICU bed with ventilators), trend study, impact on business verticals, research publications finding, guidelines for public health Safety, news, and growth projections. To the best of our knowledge, we have not found any ontology that describes all the seven dimensions D1 to D7. Nor was any such ontology found which could provide an interlink between them. All ontologies have different scopes and common goals to provide the schema for Covid-19 data.

**Table 1.** Seven dimensions covering the Covid-19 information.

| S. No | Dimension | Description | Core Classes in CovidO |
|---|---|---|---|
| 1 | **D1: Covid-19 Cases Information** | Attribute a Covid-19 case such as active, recovered, deceased, migrated cases daily across the geo-location (district, state, and country). | Statistics |
| 2 | **D2: Covid-19 Patient Information** | Represents the COVID-19 patients. Patient symptoms, suspected covid -19 cases, covid-19 treatment facility, patient travel history, patient nationality, interpersonal relationships between patients, supposed transmission reason, tracking of the patient test, etc. | Patient |
| 3 | **D3: Covid-19 Disease, Symptom and Treatment** | Describe the various diseases, different variants of disease, viruses and discover their symptom, treatment for disease. | Symptom, Disease, Treatment, Vaccine |
| 4 | **D4: Covid-19 Resources** | covid-19 clinical facility (Covid center, hospital, ambulance, available test), Availability of resources (Doctor, Nurse, Medical Equipment, Medicines etc.). Availability of bed, ICU bed with oxygen. | Resources, Covid-19 Clinical Facility |
| 5 | **D5: Covid-19 Impact on business vertical** | Impact of Covid-19 on various sectors like education sector, banking sector, economic etc. | Covid-19Impact |
| 6 | **D6: Covid-19 Related Event and Decision** | Lockdown, guideline issued by government, requirement of mask, sanitizer spray, awareness program, event hosting, various exposure of covid-19 etc. | Lockdown, Prevention, Exposure |
| 7 | **D7: Covid-19 Research Domain** | Provide the information regarding the research on Coronavirus diseases and findings. | Covid-19 Research , Covid-19 News |

# 4  The Coronavirus Disease Ontology (CovidO)

The CovidO knowledge representation model encodes knowledge in the form of classes, relationships, properties, instances, and axioms [3]. Our work is inspired by COviD-19 Ontology (CODO), and we have taken their work forward with some new features and new dimensions that cover all aspects of COVID-19. We have significantly expanded the capabilities of the codo ontology model made for Covide-19 cases and patients information, i.e., changes to classes, properties, relations, and axioms, emphasizing flexibility and extensibility. We also provide the integration by reusing existing ontology. We are giving annotations for ontology concepts that have already been produced but neither annotated nor labeled.

## 4.1  Design Methodology

From the survey of the literature one can found that there are several ontology development methodology (ODM) which carried set of acivites to create ontologies. In ODM, Knowledge acquisition, integration, and alignment drive the speed of building ontology and these comes with the risk of redundancy, consistency, and conflicts.

Invoking the existing ODM, we opted a mixed approach of Diligentand [21] and Methontology [22] development methodology to establish the CovidO. The general procedure, roles, and functions of the CovidO methodology process described as:

- **Ontology Requirement Specification (ORs):** In the ORs process, set the goal for ontology development and study the feasibility of the environment. We examine heterogeneous data sources given in the introduction to building ORs for CovidO and state-of-the-art requirement analysis according to the dimension of CovidO given in Section 3. ORs document help to design competency questions (given in Table 4) to define Covido's scope. CovidO domain knowledge should be organised as a meaningful ORs model at the knowledge level. After gathering sufficient information and ORs, we create a CovidO conceptual model that describes the problem and its solution (shown in Fig. 4).
- **Adapting Ontology:** In Diligent, the foaf: Agent starts the ontology development process by creating the initial ontology. Thus we have assumed that the initial ontology is already constructed in the form of codo. It saves ontology development time and will help expand ontology in a distributed manner with different stakeholders' objectives. Thus, ontology can have a comprehensive scope and variety of applications and needs.
- **Revision Update:** The analysis attempts to find the similarities in changing requests and users' ontology. The study looks for commonalities between the ontologies of changing requests and users. The current plan is not to combine all user ontologies. Instead, decide what changes should be made to the extended version of the shared ontology.

– **Implementation:** Once have a conceptual design, then implementation can be done with two intermdiate way: formalization and ontology design. In formalization, totransform the conceptual modelinto a formal or semi-compatible model. In the ontology design, to make ontology computable with some ontology editor like protégé with supported formal language.

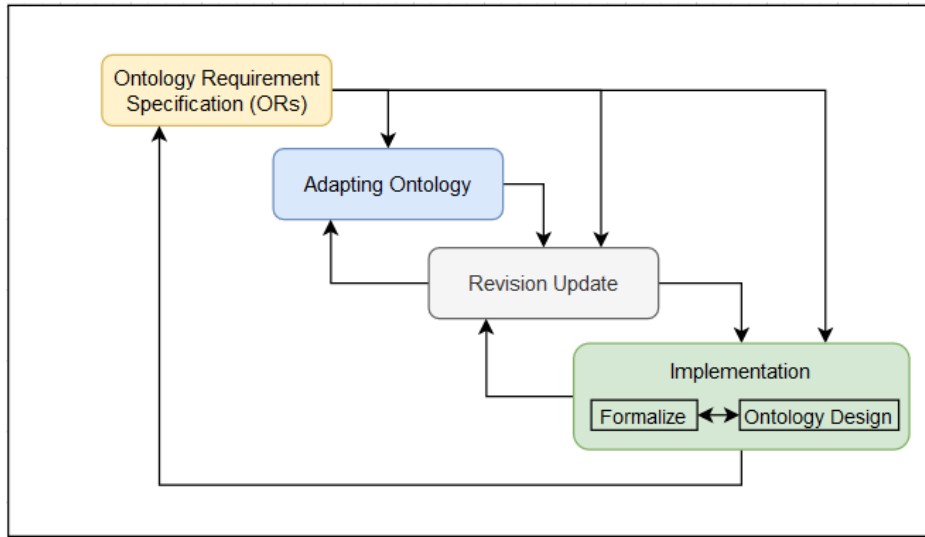

**Fig. 1.** CovidO development process methodology based on Diligentand  [21] and Methontology  [22].

## 4.2    New Features

CovidO is formed by reusing existing ontologies adding new classes and properties to cover all the dimensions listed in section 3. Table 2 represents the core classes of CovidO with referral namespace. We have divided the Statistics class of codo into two parts, codo: Statistics class and CovidO: Resource. Statistics class represents the actual cases information, and Resource class represents the resources utilized for covid-19. Similarly, other classes have been added to be meaningful, easily accessed, and used. Table 3 describes some new relationships between the classes of CovidO that were not present in other Covid ontologies. Currently, CovidO contains 175 classes, 169 relationship types, goes to evaluation. We applied the Pellet Reasoner to verify that Covido had a good consistency.

The top-level class structure diagram and relationships between core concepts are shown in Fig. 2). There are many types of relationships between concepts.

**Table 2.** Some latest added CovidO core classes with referral namespaces and their description.

| S.No | Core Class | Namespace | Description |
|---|---|---|---|
| 1. | dbo:Continent | dbpedia.org | A continent is any of several large landmasses. |
| 2. | dbo: Town | dbpedia.org | A town is a human settlement. |
| 3. | covido: TownWis-eStatistics | w3id.org/covido | Statical information of covid case in a town. |
| 4. | codo:Statistics | w3id.org/codo | update, information about the infected cases. |
| 5. | covido:Status | w3id.org/covido | provide the status as recovered, hospitalized, deceased. |
| 6. | covido:TestResult | w3id.org/covido | Provide test result positive or negative. |
| 7. | covido:Resources | w3id.org/covido | Details of Covid-19 related resourcess. |
| 8. | covido: ClinicalMeasurement | w3id.org/covido | Describe clinical measurement required for covid testing i.e. blood test, oxygen saturation, temperature etc. |
| 9. | covid19IBO: Impact | semanticweb.org/archana/ontologies/2021/5/untitled-ontology-6 | Covid-19 impact on business verticals. |
| 10. | covido: Covid-19 | w3id.org/covido | Inforamtion about the Covid-19. |
| 11. | covido: Covid-19Study | w3id.org/covido | Covid-19 study on application, news, article, global research, origin of covid etc. |
| 12. | covido: Covid19FungalInfection | w3id.org/covido | Fungal infections in covid positive patients and different variants. |
| 13. | covido: LaboratoryTest | w3id.org/covido | Laboratory Test for covid specific. |
| 14. | covido: Treatment | w3id.org/covido | A planned operation based on data for the delivery of health care. |
| 15. | covido: Vaccine | w3id.org/covido | Antigenic chemicals are used in preparations to stimulate the immune system and elicit an immunological response. |

**Table 3.** Some newly added core object properties of CovidO with domain and range.

| S.No | Object properties | Namespace | Domain | Range |
|---|---|---|---|---|
| 1. | isPartOf | w3id.org/covido | dbo:Continent | dbo:Continent |
| 2. | hasResources | w3id.org/covido | codo:Covid-19DedicatedFacility | covido:Resources |
| 3. | hasClinicalFinding | w3id.org/covido | schema:MedicalClinic | covido:ClinicalFinding |
| 4. | isVaccinated | w3id.org/covido | foaf:Person | dbo:Vaccine |
| 5. | hasImpactOn | w3id.org/covido | covido:Covid-19 | xmlns:Organization |
| 6. | hasPrevention | w3id.org/covido | covido:Mndate | covido:Prevention |
| 7. | hasDecision | w3id.org/covido | foaf:Person | covido:Decison |
| 8. | hasImpactOn | w3id.org/covido | xmlns:Organization | xmlns:Organization |
| 9. | mandatissued | w3id.org/covido | covido:PublicAuthority | covido:Mndate |
| 10. | hasRiskFactor | w3id.org/covido | xmlns:Organization | covido:RiskFacotr |
| 11. | hasGovern | w3id.org/covido | covido:PublicAuthority | xmlns:Organization |

The content ensures the semantic consistency of relationships between concepts and facilitates logical axioms and reasoning definitions.

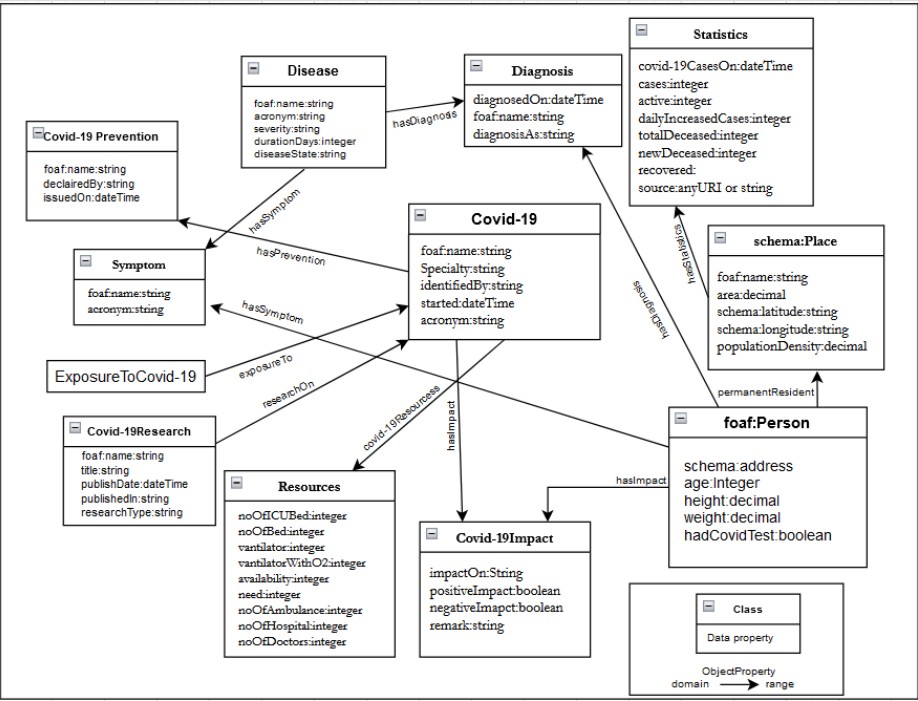

**Fig. 2.** class structure diagram of the top levels of the core class hierarchy in CovidO.

### 4.3   CovidO Scope

Competency Questions(CQs) play an essential role in the ontology development lifecycle, representing the ontology requirements. Some recommendations have to formalize requirements through Covid-19 data sources.We could structure and expand the conceptual modeling design of codo to enhance codo model capabilities using competency questions (CQ:s), reasoning requirements, and contextual statements. The competency questions have broad coverage on Covid-19 data, so we are trying to reduce it and map into the seven dimensions of CovidO to cover all the aspects of the pandemic. Table 4 represents some competency questions with the respective dimension that CovidO is expected to answer.

### 4.4   Reusing the ontology concept

According to  [18], there are two perspectives to reuse ontology: (1) assembling, extending, specializing, and adapting other ontologies that are components of

**Table 4.** CovidO competency questions excerpt.

| Dimension | Q.No. | Competency Questions |
|---|---|---|
| D1 | CQ1. | When did country c have the first Covid-19 case? |
|  | CQ2. | How many Covid-19 cases have been found at any (location) place l on date t? |
|  | CQ3. | How many patients died in Continent r on date t? |
| D2 | CQ4. | What is the travel history of a patient p? |
|  | CQ5. | How many patients travelled from / to Continent r on date t? |
|  | CQ6. | List all the patients between age 18 and 30. |
| D3 | CQ7. | What are the different variants of Covid-19? |
|  | CQ8. | What are the prevention vaccines for Covid-19? |
|  | CQ9. | What are the drugs for Covid-19 treatment? |
|  | CQ10. | What are the clinical measurements of a Covid-19 patient p? |
| D4 | CQ11. | What are the health care facilities for Covid-19 in a place l? |
|  | CQ12. | How many ICU beds are in a hospital h at a place p? |
|  | CQ13. | How many ambulances are available in a hospital h on date t? |
| D5 | CQ14. | What are the business verticals on which COVID-19 has a positive impact? |
|  | CQ15. | Provide a list of all organizations at high risk? |
| D6 | CQ16. | When was the first lock down announced in a country c? |
|  | CQ17. | What is the exposure of Covid-19 spread? |
|  | CQ18. | What is prevention advice issued by state public authority for covid-19? |
| D7 | CQ19. | What are the research articles published for Covid-19 in a month? |
|  | CQ20. | What are the news headlines and their sources of covid-19 on date t? |

the final ontology, or (2) integrating other ontologies on the single concept that integrate all concept. We have used second approach, integration of other ontologies. The core concepts for CovidO are determined as prevention, vaccine, hospital facility, disease, infection, disorder, virus, SARS-Cov2, Coronavirus, agent, patient, disease, symptom, drug, treatment, organization, impact, host, diagnosis, statistics, place, etc. Which is already in use in other ontologies O1 to O12. CovidO integrate the concept with existing ontologies to pursue the basic principle of ontology implementation. Fig. 3 represents the terms inherited by CovidO from available schemas. Different colors define each schema. The solid and the dotted line show immediate and remote child-parent relations between classes. Most inherited entities belong to Schema, BFO, and CODO, while Kg4Grug, SYMP, and VO are the least inherited ontologies.

## 4.5   Conceptual Design of CovidO

We use a permanent URL service, w3id.org, that makes it independent of external ontologies. We have defined a new namespace convention https://w3id.org/covido for all classes used in CovidO, with the prefix covido (the entry registered at http://prefix.cc ). A common concept is integrated into a single concept that unifies them all. The concept selection is based on the best fit for the scope and dimension of CovidO. CovidO relevant file, competency questions, data, code, and owl file have been provided by the GitHub community, advantaging

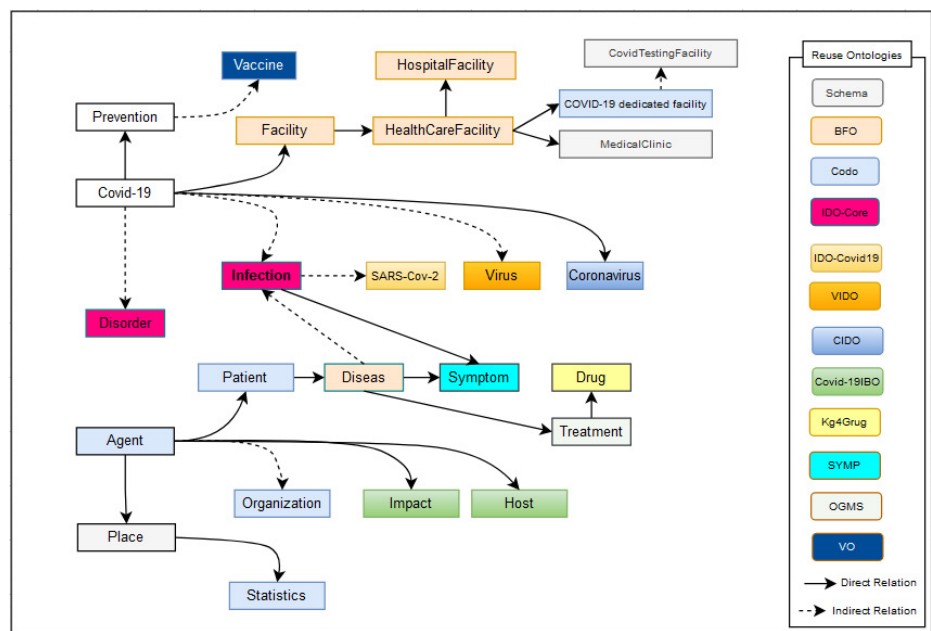

**Fig. 3.** Reuse core conpets of CovidO ontology extracted from the related ontologies.

to the contributor. A overview of classes and properties of CovidO is represented in Fig. 4. Complete CovidO structure has seven dimensions (Disease and Treatment, Resources, Cases, Patient, Events, Research, and Symptoms), seven dimension have been represented in different collor. Most of the entities in our ontology belong to the Disease and Treatment dimension. The resources dimension is the smallest in terms of the number of classes. The solid and the dotted arrow represents the property and the subclass relation between two entities respectively. The name over the arrow represents the object property, the arrow tail connects the domain, and the head is the range of property. Vertical containers represent the class with inside data properties. The data properties follow up vocabularynamespace:dataproperty:datatype.

## 5    Evaluation

We evaluated CovidO in two ways: using SPARQL query and OOPS! Pitfall Scanner. SPARQL query describes the accessibility of elements and OOPS! Pitfall represents the RDF quality of CovidO.

### 5.1    SPARQL Query Evaluation

SPARQL [19] allows users to query necessary and alternate graph patterns and their conjunctions and disjunctions. CovidO is schema-agnostic, so schema

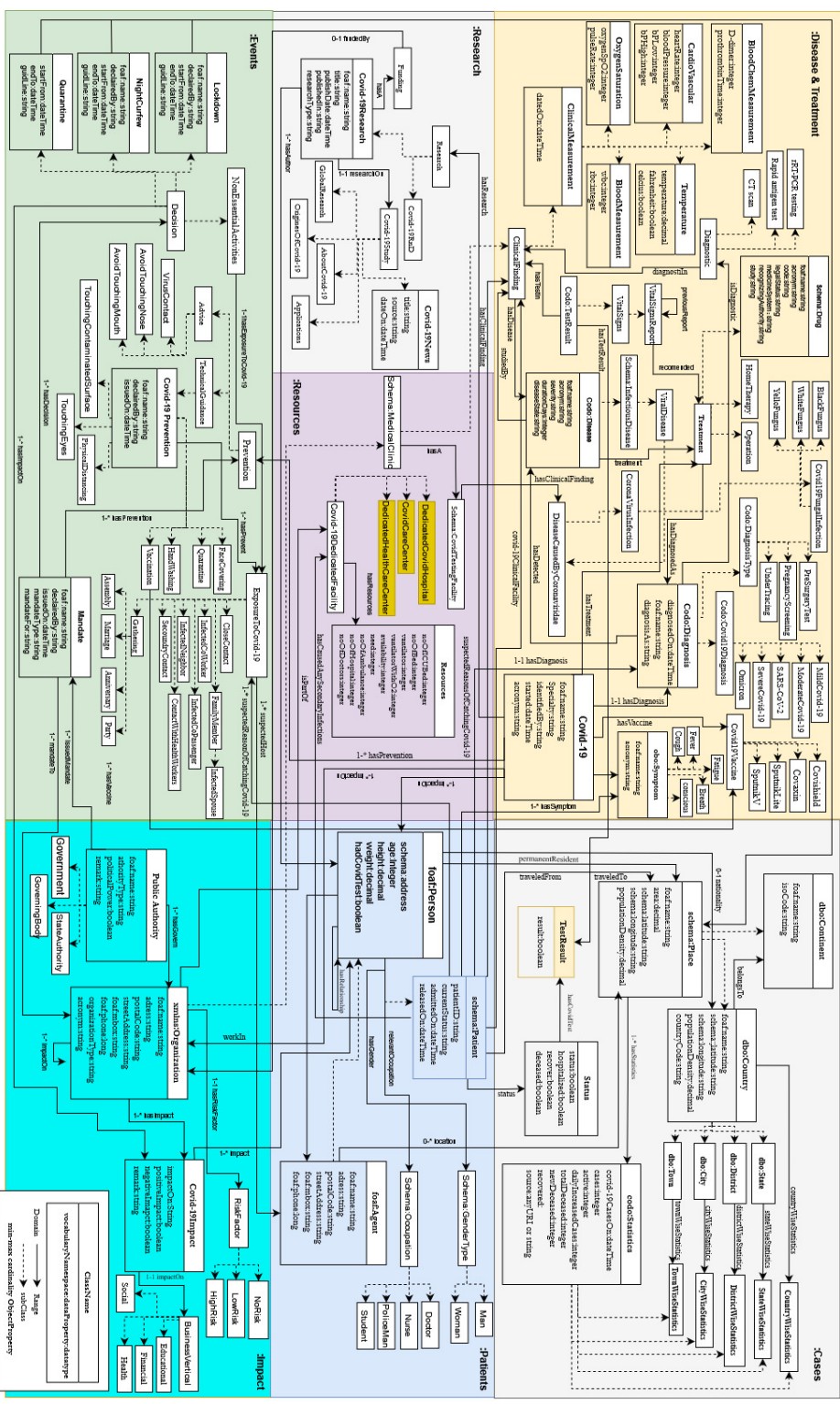

**Fig. 4.** A figure represent the conceptual design of CovidO.

level SPARQL query is allowed. Once the ordered representation is done as user requirement from the coronavirus distributed and heterogeneous data sources, CovidO instance-level questions are permitted to answer. We evaluate CovidO on schema-level competency questions based on the D3 dimension. The SPARQL queries and their results are shown in Table 5. The prefix and their namespaces used for SPARQL queries are shown in Table 6.

**Table 5.** CovidO schema level queries and their SPARQL queries and results obtained.

| S.No | Question | SPARQL Query | Result |
|------|----------|--------------|--------|
| CQ8. | What are the prevention vaccines for Covid-19? | SELECT ?CovidVaccine WHERE { ?Vaccine rdfs:subClassOf covido:Vaccine. ?CovidVaccine rdfs:subClassOf ?Vaccine.} | Novavax, Covaxin, Moderna, SputnikLite, Zydus-Cadila, Covishield, Corbevax, SputnikV, |
| CQ10. | What are the clinical measurements of a Covid-19 patient? | SELECT * WHERE ?Vaccine rdfs:subClassOf covido:ClinicalMeasurement. | Temperature, OxygenSaturation, BloodMeasurement, CardioVascular, BloodChemMeasurement, Diabetes |

**Table 6.** CovidO prefixes table.

| Prefix | Namespace |
|--------|-----------|
| rdf: | http://www.w3.org/1999/02/22-rdf-syntax-ns |
| owl: | http://www.w3.org/2002/07/owl |
| rdfs: | http://www.w3.org/2000/01/rdf-schema |
| xsd: | http://www.w3.org/2001/XMLSchema |
| foaf: | http://xmlns.com/foaf/0.1/ |
| ndf: | https://www.semintelligence.org/ns/whonto |
| xmlns: | http://xmlns.com/foaf/0.1/ |
| dbo: | http://dbpedia.org/ontology/ |
| schema: | https://schema.org/ |
| covido: | http://www.w3id.org/covido/ |

## 5.2   OOPS! Pitfall Evaluation

We have used OOPS! Pitfall scanner  [20] to examine CovidO. OOPS! Pitfall ontology diagnosis online tool detects 40 different types of pitfalls in OWL ontologies, including semantic and structural checks and best practices verification. OOPS! Pitfall scanner has two components, Pitfall Scanner, and Suggestion Scanner. Pitfall Scanner checked the ontology syntax, and Pellet reasoner has

analyzed the logical consistency of ontology. Whereas the Suggestion scanner has thrown some suggestions for possible errors of ontology elements. OOPS! Pitfall based result evaluation on CovidO is shown in the Fig. 5. We resolved the problems reported by OOPS! in the CovidO and push the changes to the target GitHub repository.

**Fig. 5.** Screenshot of result summary of CovidO on OOPS! Pitfall Scanner.

## 6    Conclusion

This work presents Coronavirus Disease Ontology (CovidO): 1) Describes complete knowledge of Covid-19; 2) Maps existing ontologies and create standard metadata to understand and share Covid-19 knowledge that humans perform to interact with the same affordances; 3) Acts as a vocabulary for researchers, engineers, and developers to search/identify the commonly used Covid-19 gesticulation for particular affordances, and to understand the scope and dynamics of a specific gesture. We extracted elements observed from existing coronavirus ontologies defined in previous studies while developing the ontology. The intention and scope of the CovidO can be summarized in seven dimensions as discussed above.

As future work, several possible extensions can be made to the ontology by integrating linguistics and adding more uses case. In addition, we intend to release and deploy the CovidO RESTful service in the Cloud and API clients to the leading coronavirus annotation.

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
