# OpenReview forum: "The Coronavirus Disease Ontology (CovidO)"
_kg-construct.github.io/KGCW/2022/Workshop — Submitted to KGCW 2022_

### Official Review · ~Giorgos_Flouris2 · 2022-03-21
**A useful and important work, but poorly presented**

**Rating:** 4
**Confidence:** 4

**Review:**

This paper presents an ontology developed by the authors, which aims to organise information regarding the COVID-19 pandemic and associated issues. The authors reuse several existing ontologies on the subject in order to create one "unifying" schema that allows the representation of several different aspects of the pandemic (financial, statistical, medical etc.).

Although the work seems well-motivated and useful, the paper seems hastily written and does not fully convey the message. As such, it does not serve its purpose as an introductory documentation for the ontology, and thus is not useful for the interested reader, who will not be sufficiently informed about the capabilities and structure of the ontology just by reading the paper.

More specifically, there are several sentences in the paper that simply make no sense, as well as a significant number of typos (see below for more details). Moreover, although the necessary sections or subsections that are required to present the work are there (e.g., methodology, competency queries, conceptual design etc.), these sections/subsections do not adequately communicate what they are supposed to. Unfortunately, it is not just an issue of poor english, it is more a problem of giving inadequate details for each subject, eating up space with generic statements that do not enlighten the reader.

Some further comments:

- It would be good to mention whether the provided structure (concepts/properties of the ontology) would be useful for other types of diseases or viruses (e.g., flu). An estimate on the amount of work required to adapt the ontology for such alternative uses would be useful.

- On the related work: O1, O2, O3 and O4 are essentially the same ontology, gradually specialised. Why is it presented as 4 different ontologies?

- "CovidO relevant file, competency questions, data, code, and owl file have been provided by the GitHub community, advantaging to the contributor". What does this mean? Did someone else develop the ontology?

- On the evaluation: the use of the OOPS! Pitfall ontology diagnosis tool is a good idea. However, I don't understand why Fig. 5 is shown. The authors supposedly resolved those problems, right? They were valid in some preliminary version of the ontology, but now they are resolved, right? If so, why is Fig. 5 presented?
Also, I'm not sure what Table 5 proves. OK, the ontology can answer some SPARQL queries, but what does this mean? Why is this important for the evaluation?

Typos and other presentation issues (partial list):

- Covide -> covid (in several positions)

- "have been introduced required"

- "communityand"

- "drive the speed" ?

- "and these comes"

- What is "revision update"? (third bullet in Subsection 4.1)

- "We applied the Pellet Reasoner to verify that Covido had a good consistency." -> "We applied the Pellet Reasoner to verify that Covido is consistent."

- "to query necessary and alternate graph patterns" ??

Examples of sentences that make no sense (partial list):

- "From the survey of the literature one can found that there are several ontology development methodology (ODM) which carried set of acivites to create ontologies."

- "In Diligent, the foaf: Agent starts the ontology development process by creating the initial ontology"

- "Once have a conceptual design, then implementation can be done with two intermdiate way: formalization and ontology design. In formalization, totransform the conceptual modelinto a formal or semicompatible model. In the ontology design, to make ontology computable with some ontology editor like protégé with supported formal language."

- "Currently, CovidO contains 175 classes, 169 relationship types, goes to
evaluation."

- "Once the ordered representation is done as user requirement from the coronavirus distributed and heterogeneous data sources, CovidO instance-level questions are permitted to answer."

---

### Official Review · ~Ana_Iglesias-Molina2 · 2022-03-27
**Covid ontology yet improvable**

**Rating:** 3
**Confidence:** 5

**Review:**

This paper presents CovidO, an ontology that aims to represent the different aspects of Covid-19 related data present in other published ontologies related to the mentioned disease. It has been developed and published following the Diligent and Methontology methodologies for ontology development, implemented, published with a persistent IRI and evaluated with OOPS! and SPARQL queries.

The paper has a good motivation, but it needs refinement in several aspects. The ontology has been built upon an analysis of the dimensions of Covid-19 information, but these dimensions are poorly described. They are presented, but there is no argumentation of how and why those specific ones, not even some correlation between the ontologies presented in the related work and the dimensions. Thus, the paper cannot really prove that the presented ontology covers “all the possible dimensions”, as strongly and often claimed in the paper, neither that the presented dimensions of covid-19 data are complete.

Regarding methodology and the resources available of the ontology I have several concerns. The ontology is published online with an HTML documentation and the link is provided. The documentation itself lacks a proper description, diagrams and examples to be complete and comprehensible. I expect to find that in an ontology documentation, or in the GitHub repository. The link to the mentioned GitHub repository is not provided in the paper, and the one I found ([https://github.com/sumitsnit/CovidO](https://github.com/sumitsnit/CovidO)) only provides basic information. Thus, I could not find the requirements specification, including the complete set of competency questions and SPARQL queries, so the results provided in the paper cannot be reproduced. The evaluation with OOPS! turned out surprising, because the pitfall report that I obtained using the ontology URI returned a critical error (Results for P37: Ontology not available on the Web); and with RDF, it outputs almost the same minor and important errors that the authors claimed had been solved. The results of the Pellet reasoner were fine.

Additional minor comments:
* The writing of the paper could greatly improve, there are several sentences that make no sense or not understandable; missing spaces to separate words more often than expected (even for names, Section 4.1 second paragraph: “Diligentand” instead of “Diligent”), typos, etc.
* Tables 2 and 3 seem expendable, it is not clear what authors are trying to show with them
* Figure 4 is really hard to read, it would gain in understandability if it was divided and each dimension presented separately as a module.
* The paper could use footnotes to add the links that are in the text

In general, the purpose of the ontology is fine, but the resources of an ontology of this size and potential needs further development to address the issues mentioned above, along with the writing of the paper. Furthermore, since the ontology provided is not populated and the associated KG is missing, I think there are more suitable venues for an ontology description paper like this one.

---

### Official Review · ~Franck_Michel1 · 2022-03-30
**Probably a good work but hardly verifiable and off scope for KGCW**

**Rating:** 3
**Confidence:** 5

**Review:**

This paper presents the Coronavirus Disease Ontology (CovidO) that gathers, interconnects and extends multiple existing ontologies to cover a broad range of use cases wrt. the Covid19-related information, split along 7 main domains.

Overall the work seems interesting and certainly required a substantial work. However in the current status, the paper makes it hard to assess the quality of this work and the resulting ontology.

Furthermore, the paper seems totally off scope wrt. to the goals of the workshop: "KGCW22 has a special focus this time on the automatization of knowledge graph construction methods, analyzing their alignment with previous standard but declarative approaches (i.e., mapping rules)."
The work reports consists of a - probably manual - construction of an ontology.

The paper states that CovidO covers ALL of the concepts/attributes/features/relations needed to describe anything related to Covid19. This is a very strong claim that is hardly verifiable.
Competency questions in Table 4 are related to one domain whereas most of them actually cover 2 domains. E.g. CQ4 covers domains Patient and Cases (for the travel history).
But beyond this, there is a lack for CQs that would illustrate the whole CovidO need, that is, involve multiple domains such that CovidO makes it possible to solve questions that are out of reach for previous works.

The github repo is not given. I found this one https://github.com/sumitsnit/CovidO which is mostly empty.
This is very disappointing given that it is supposed to provide all the code, CQs, OWL files etc. as mentioned in §4.5.
Furthermore, at the time of reading, URL http://w3id.org/covido returns an error 404 such I could never see the ontology anywhere.

Many sentences are vague, unclear, confusing, senseless or even wrong. Examples:
- "Covid is schema agnostic"
- "Some recommendations have to formalize requirements through Covid-19 data sources"
- about the ROCS: "study the feasibility of the environment": which environment? Feasibility wrt. what?
- "CovidO domain knowledge should be organized as a meaningful ORs model at the knowledge level", what does this even mean?
- "The study looks for commonalities between the ontologies of changing requests and users."
- "formal or semi-compatible model": what dies this mean?
What does this even mean??
In addition, multiple typos make the whole reading pretty tough.

The methodology (Section 4.1) consists of rather confused general considerations, and it does not convince the reader about whether the methodology has really been applied. And multiple sentences are very unclear (see above).
The implementation mentions "ontology design" as one way of implementing it whereas it usually refers to the whole methodology.
This section also contains lots of typos.

FOAF is used with 2 seperate prefixes, foaf and xmlns. The latter is useless and confusing.

The related works could mention the Schema.org additions for Covid19 (http://blog.schema.org/2020/03/schema-for-coronavirus-special.html). Possibly these could be reused.

---

### Official Review · ~Sven_Lieber1 · 2022-03-31
**Interesting and important work, yet the presentation needs to be improved**

**Rating:** 3
**Confidence:** 4

**Review:**


This paper presents the CovidO ontology with the aim to combine several other Covid-19 ontologies covering only limited scopes.
Whereas the work is well motivated, the presentation of the paper can be improved.

The work is well motivated and the development process seemed to have followed common best practices
- A persistent URI was chosen using w3id, the prefix registered at prefix.cc
- competency questions were created and are publicly available vai GitHub (at least this is mentioned)
- existing ontologies are reused
- the ontology was developed by following an existing ontology development methodology

However, the GitHub link is not provided, only mentioned.
Online I found the following GitHub repository, but unfortunately it does not contain the promised content: https://github.com/sumitsnit/CovidO

The paper was not easy to follow and the presentation of the paper can be improved (see below).
This makes it hard to judge the quality of the work.

Regarding the content I am curious about the following to aspects

- A lot of different ontologies were reused, even though a consistency check was performed with the Pellet reasoner, I am interested in the authors approach to assess existing axioms
and ensure consistency. How time consuming was this process? Were any problems encountered by trying to combine the different ontologies which possibly influenced modeling decisions of CovidO?

- This workshop is about Knowledge Graph generation, were any automated methods used to create the ontology?

The conclusion states that complete knowledge of Covid-19 is described. This is a pretty strong statement, how can it be complete? You mention yourself that there is future work to add more use cases

This seems like very relevant work, I therefore encourage the authors to reflect on feedback and improve the presentation of the paper, for example by fixing grammar mistakes resulting in unclear statements (see below).


## Typos/grammar complicating the interpretation:

- "They comprehensively and thoroughly describe coronavirus disease"  => the coronavirus disease or several coronavirus-related diseases?
- Regarding the outline of the paper, What are "results for the schema", results with respect to what?
- "in Diligent, the foaf: Agent starts the ontology development process" => what is a foaf Agent in this content? Do you refer to the ontology engineer?
- "other classes have been added to be meaningful, easily accessed, and used." => how was "the meaning" affected? classes have been added to be used, that seems obvious.
- "We applied the Pellet Reasoner to verify that Covido had a good consistency." => What is a good consistency? Is it not always consistent but mostly? In which cases isnt'the ontology consistent according to the reasoner?
- "CovidO is schema-agnostic, so schema level SPARQL query is allowed. Once the ordered representation is done as user requirement from the coronavirus distribuured and heterogeneous data sources, CovidO instance-level questions are permitted to answer." => This sentence is not clear to me, does it mean that with SPARQL the classes and relationships can be queried too? How does this affect the instance level questions? What are permissions for the answers?
- Consistency: coronavirus, Covid-19, ...

## Typos/grammar minor:

- "The contents of this paper are organized as follows" => The content of this paper is organized as follows
- "Another vital issue of Covid-19 is to providesemantic (machine understandable) representation of data from various exciting felds such as research, health, resources, drugs, and treatment" => existing?
- Several typos which could have been easily identified fixed using grammar-check funcationality of standard text editing tools: "totransform", "modelinto"
- several times: ontology => ontologies

---

### Decision · Program_Chairs · 2022-04-11

**Decision:**

Reject

**Comment:**

Dear authors,

Thank you for submitting your paper. Unfortunately we don’t accept your paper now in its current state. We refer to the reviews for suggestions on how you can improve your paper.

Kind regards
Organizers of the Knowledge Graph Construction workshop 2022